# Does Dietary-Induced Obesity in Old Age Impair the Contractile Performance of Isolated Mouse Soleus, Extensor Digitorum Longus and Diaphragm Skeletal Muscles?

**DOI:** 10.3390/nu11030505

**Published:** 2019-02-27

**Authors:** Cameron Hill, Rob S. James, Val M. Cox, Jason Tallis

**Affiliations:** Centre for Sport, Exercise and Life Sciences, Alison Gingell Building, Coventry University, Priory Street, CV1 5FB Coventry, UK; apx214@coventry.ac.uk (R.S.J.); apx253@coventry.ac.uk (V.M.C.); tallisj2@uni.coventry.ac.uk (J.T.)

**Keywords:** ageing, obesity, muscle, power, work loop

## Abstract

Ageing and obesity independently have been shown to significantly impair isolated muscle contractile properties, though their synergistic effects are poorly understood. We uniquely examined the effects of 9 weeks of a high-fat diet (HFD) on isometric force, work loop power output (PO) across a range of contractile velocities, and fatigability of 79-week-old soleus, extensor digitorum longus (EDL) and diaphragm compared with age-matched lean controls. The dietary intervention resulted in a significant increase in body mass and gonadal fat pad mass compared to the control group. Despite increased muscle mass for HFD soleus and EDL, absolute isometric force, isometric stress (force/CSA), PO normalised to muscle mass and fatigability was unchanged, although absolute PO was significantly greater. Obesity did not cause an alteration in the contractile velocity that elicited maximal PO. In the obese group, normalised diaphragm PO was significantly reduced, with a tendency for reduced isometric stress and fatigability was unchanged. HFD soleus isolated from larger animals produced lower maximal PO which may relate to impaired balance in older, larger adults. The increase in absolute PO is smaller than the magnitude of weight gain, meaning in vivo locomotor function is likely to be impaired in old obese adults, with an association between greater body mass and poorer normalised power output for the soleus. An obesity-induced reduction in diaphragm contractility will likely impair in vivo respiratory function and consequently contribute further to the negative cycle of obesity.

## 1. Introduction

Older adults are reported to experience poorer muscular strength and power output (PO) resulting in reduced locomotory function and quality of life [1]. It is suggested that obesity in old age may exacerbate these effects [2], however, whilst obesity in young adults is known to significantly alter skeletal muscle function [3,4,5], the possible synergism between the effects of old age and obesity on muscle function is poorly understood. The limited available evidence examining the impact of obesity on the muscle function of older adults is equivocal, with evidence demonstrating either no change [4,6], or an increase in the absolute force of the lower leg musculature [7], whilst others have shown a reduction in plantar flexor and dorsiflexor absolute force and power and force relative to body mass [8]. A recent review has indicated that assessing the effect of obesity using an isolated muscle model can help further our understanding of obesity effects on contractile performance [2]. Such models have been used regularly to examine skeletal muscle ageing and more recently, obesity effects on isolated muscle contractility, though a distinct lack of literature has examined their concomitant effects on isolated muscle function [2].

Evidence indicates that obesity and old age independently result in a reduction in muscle quality (force or PO relative to muscle size), which results in the formation of larger muscles of lower quality [9,10]. For obese individuals particularly, such effects may contribute to an already elevated body mass for the same, or lower, mechanical work [9]. Measures of muscle quality are difficult to ascertain in vivo, with absolute changes in strength and PO commonly reported for old obese adults [4,6,7,8,11]. Whilst work has normalised contractile performance to body mass [8] and muscle volume [12], these approaches fail to consider whole tissue mass that can be otherwise obtained via in vitro examinations of whole skeletal muscles [2].

Mechanistically, poorer muscle function in older animals and humans has been attributed to impaired calcium handling, reduced contractile mass, impaired cross-bridge kinetics, a shift in fibre type composition, greater lipid accumulation and chronic inflammation [2]. In obesity models, mechanisms causing a decline in contractile function are similar to the mechanisms that cause a decline in contractile function in old age [13,14,15]. As a consequence, isometric stress (force relative to muscle cross-sectional area) and PO normalised to muscle mass is impaired in aged [16,17,18,19,20] and young obese [9,21,22,23] isolated skeletal muscles. Such mechanisms form the basis for an obesity-associated, muscle-specific reduction in isolated muscle contractile performance in old, obese mammalian muscles.

We aimed to uniquely determine whether dietary-induced obesity in old age worsens skeletal muscle contractile function at the isolated muscle level via usage of the work loop technique. Previous isolated muscle studies comparing contractile function in old age and in dietary-induced obesity, independently examine isometric force [18,20,21,22,23] whereas the work loop technique considers the interactions between force production during shortening, the force-velocity relationship, and the passive work required to lengthening the muscle, providing a better representation of in vivo muscle function [24,25]. Previous work from our lab using the work loop technique has demonstrated that isolated extensor digitorum longus (EDL) and diaphragm isometric stress and work loop PO normalised to muscle mass declines by 50 weeks [16] and 78 weeks of age [17]. In addition, we have also examined the effects of dietary-induced obesity on isolated soleus, EDL and diaphragm isometric force and work loop power output in relatively young (20–30 weeks) female CD-1 mice fed a high-fat diet (HFD) for 16 weeks [9] and for durations of 2, 4, 8 and 12 weeks [26], compared to age-matched control animals in each study. As our lab has previously examined the effects of dietary-induced obesity on contractile function in young animals, with no study to date having examined dietary-induced obesity in old age, the current work examines changes in isolated locomotory (soleus & EDL) and respiratory (diaphragm) contractile function in old (79 weeks) female mice. It is proposed that dietary-induced obesity in old age will cause muscle-specific reduction in isometric stress, work loop PO normalised to muscle mass and fatigue resistance, given the similar mechanistic adaptations of skeletal muscles to ageing and obesity

## 2. Materials and Methods

### 2.1. Animals

Ethics approval was provided by the Coventry University ethics committee prior to the commencement of this study (P36339; 15/10/2015). 60 female mice (strain CD-1, Charles River, UK) were purchased at 9 weeks old and matured in-house in groups of 8–10 at Coventry University without access to running wheels. Animals were kept in a 12:12-hour light:dark cycle at 50% humidity. Access to standard lab chow (CRM(P); SDS/Dietex International Ltd, Whitham, UK; calories provided by protein 17.49%, fat 7.42%, carbohydrate, 75.09%; gross energy 3.52 kcal/g; metabolizable energy 2.57 kcal/g) and water was provided ad libitum. At 68 weeks of age, mice were divided into cages of 10 and were assigned to either a control (*n* = 30) or high-fat diet (HFD; *n* = 30) group, ensuring each group was matched for body mass (Control, 49.8 ± 1.2 g; HFD, 49.8 ± 1.3 g; mean ± standard error of mean [S.E.M.]; *P* = 0.99).

Animals acclimated to their new groups for 2 weeks, with each dietary protocol commencing at 70 weeks of age for a duration of 9 weeks. This feeding duration was selected as previous work has demonstrated substantial changes in body composition and muscle mechanics following a similar feeding duration of the same dietary sources in younger CD-1 female mice [26], therefore, ensuring substantial alterations in body fatness between the two experimental groups. Moreover, by commencing the dietary protocols at 70 weeks of age, this approach allowed for substantial ageing to occur prior to the induction of obesity, thus allowing for a clear examination of the synergism between obesity and old age. The control group were provided the standard lab chow whilst the HFD group were provided with a self-selected forage diet of husked sunflower seeds (Advanced Protocol PicoLab, Fort Worth, USA; calories provided by protein 17.95%, fat 63.66%, carbohydrate, 18.39%; gross energy 5.24 kcal/g; metabolizable energy 3.80 kcal/g) in addition to the standard chow. Both dietary groups had ad libitum to each diet and water for the feeding duration. A 79-week-old age group was chosen for contractility measures as 50% survivorship of female CD-1 mice is 78–80 weeks [27], and significant reductions in soleus and EDL isometric stress (20% and 19% respectively; *P* < 0.05) and work loop PO normalised to muscle mass (21% and 16% respectively; *P* < 0.05) have been observed by 78 weeks of age [17].

### 2.2. Muscle Preparation

Animals were sacrificed by cervical dislocation at 79 weeks of age in accordance with British Home Office Animals (Scientific Procedures) Act 1986, Schedule 1. Following sacrifice, animals were weighed to determine body mass, with nasoanal length measured using digital calipers (Fisher Scientific™ 3417, Fisher Scientific, Loughborough, UK). Body circumference around the abdomen was measured with a textiles tape measure. Body mass and nasoanal length were used to calculate Body Mass Index (Equation (1)) [28] and Lee Index of Obesity (Equation (2)) [29] for each individual. Gonadal fat pad mass was dissected and weighed to determine the differences in fat accumulation in response to each diet.
(1)$$\mathrm{Body}\;\mathrm{Mass}\;\mathrm{Index}=\frac{\mathrm{BM}\left[g\right]}{\left(\mathrm{NAL}{\left[\mathrm{cm}\right]}^{2}\right)}/100$$

Equation (1)—The equation used to calculate the Body Mass Index of a rodent [28]. Body mass, BM; Nasoanal length, NAL.
(2)$$\mathrm{Lee}\;\mathrm{Index}\;\mathrm{of}\;\mathrm{Obesity}=\frac{\sqrt[{3}]{\mathrm{BM}\left[g\right]}}{\mathrm{NAL}\left[\mathrm{cm}\right]}x\;100$$

Equation (2)—The equation used to calculate the Lee Index of Obesity of a rodent [29]. Body mass, BM; Nasoanal length, NAL.

Whole EDL or soleus (*n* = 10 per muscle per group) was dissected from the right hind limb. The proximal tendon was secured via aluminium foil T-clips and a piece of bone left at the distal tendon. A ventral segment of the costal diaphragm (*n* = 10 per group) was isolated from the right-hand portion of the rib cage, and aluminium foil T-clips wrapped around the central tendon of the diaphragm segment, with the two ribs anchoring the muscle left intact. All dissections were performed in chilled (~5 °C), oxygenated (95% O_2_, 5% CO_2_) Krebs-Henseleit solution of composition (mM) NaCl 118; KCl 4.75; MgSO_4_ 1.18; NaHCO_3_ 24.8; KH_2_PO_4_ 1.18; glucose 10; CaCl_2_ 2.54; pH 7.55 at room temperature before oxygenation.

### 2.3. Assessment of Contractile Performance

Each muscle was placed in a flow-through chamber circulated with oxygenated Krebs-Henseleit solution maintained at 37.0 ± 0.2 °C. At one end, the muscle was attached, via crocodile clips, to a force transducer (UF1, Pioden Controls Ltd, Henwood Ashford, UK) and a motor arm at the opposing end (V201, Ling Dynamic Systems, Royston, UK). The position of the motor arm was detected by a Linear Variable Displacement Transducer (DFG5.0, Solartron Metrology, Bognor Regis, UK). Muscle activation was elicited via parallel platinum electrodes, with electrical currents provided by an external power supply (PL320, Thurlby Thandar Instruments, Huntingdon, UK). Stimulation and length change parameters were altered via a custom written programme (CEC Testpoint, Measurement Computing, Norton, MA). Initially, isometric contractions were used to examine the maximal twitch and tetanus kinetics of isolated skeletal muscles as with previous studies of mammalian skeletal muscles [9,16,17,25,30] increasing physical muscle length, stimulation amplitude (12 V–14 V for soleus and diaphragm, 14 V–18 V for EDL; fixed current of 120 mA, fixed pulse width of 1.2 ms), and stimulation frequency (120 Hz–140 Hz for soleus and diaphragm, 200 Hz–220 Hz for EDL) until maximal tetanic force was achieved. Optimal length (*L*_0_) was measured using an eyepiece graticule fitted to a microscope, with estimates of mean muscle fibre length measured as 85% of physical soleus length and 75% for EDL [9]. No estimates of mean fibre length for diaphragm are available, so physical length was used as *L*_0_ [9].

Using the stimulation frequency, stimulation amplitude, and *L*_0_ which yielded maximal tetanic force, work loop PO was measured by subjecting each muscle to symmetrical sinusoidal length changes around its optimal length whilst being stimulated to produce force during shortening. PO was determined across a range of cycle frequencies (CF), or contractile velocities, in order to produce a power output-cycle frequency curve [31], which allowed us to determine whether there was a shift in the optimal CF to produce power following a HFD and whether PO changed at fast and slow CF’s. The CF’s tested ranged from 2–10 Hz for soleus, 4–18 Hz for EDL, and 3–12 Hz for diaphragm. Strain (length change amplitude) and burst duration (electrical stimulus duration) were altered at each CF for each muscle to ensure peak net work production. Electrical stimulations were controlled via a data acquisition board (KUSB3116, Keithley Instruments, Ohio, USA) and altered via a custom-designed Testpoint programme (CEC Testpoint version 7, Measurement Computing, Norton, MA, USA). Force and distance were sampled throughout the work loop cycle using a sampling rate of 10kHz and plotted against each other to produce a work loop. Net work was calculated as the positive work produced during shortening minus the work required to lengthen the muscle.

Initially, a CF of 10 Hz for EDL, 7 Hz for diaphragm and 5 Hz for soleus was used as these CF’s typically elicited maximal PO in previous research for 78-week-old locomotor [17] and young respiratory [9,16] skeletal muscles. A strain of 0.10 was employed at the aforementioned CF’s for each muscle, with a strain of 0.10 equating to a ±5% change in length from *L*_0_, i.e., the muscles lengthened by 5%, shortened by 10%, then re-lengthened by 5% back to *L*_0_. At these CF’s, phasic bursts of electrical stimulation were provided for durations of 50 ms, 55 ms and 65 ms to the EDL, diaphragm and soleus respectively. A stimulus phase (time delay for electrical stimulation) of −2 ms, −5 ms and −10 ms were utilised for EDL, diaphragm and soleus respectively. These timings ensured the stimulation was provided just prior to the muscle reaching its maximal length to ensure a rise in force before shortening. The aforementioned values for strain, stimulus phase, and burst duration have been shown to elicit maximal work loop PO at these CF’s [9,16,17,25] though was altered on an individual basis to ensure maximal work loop PO. For the other CF’s, as CF increased, strain and burst duration increased also, and vice versa. Preliminary work showed altering stimulus phase at each CF had no significant effect on net work. Each muscle was subjected to four work loops per run every 5-min to allow for sufficient recovery.

Control sets of work loops were performed using the parameters that elicited maximal net work (soleus, 5 Hz; EDL, 10 Hz; diaphragm, 7 Hz) every 3 to 4 sets of work loops, and following examination of net work for the final CF of each muscle, to monitor changes in net work over the course of the experiment. Any variation in net work was due to an impairment in force production. Therefore, the PO produced by each muscle at each CF prior to the fatigue run was corrected to the control run that yielded the greatest net work, assuming that alterations in PO were linear over time [30].

Each muscle underwent 10-min of rest prior to the fatigue run. To determine fatigability, each muscle was subjected to fifty consecutive work loops using the optimised stimulation parameters at 5 Hz, 7 Hz and 10 Hz for soleus, EDL and diaphragm respectively. The net work of every second work loop was plotted against time until each muscle produced <50% of the pre-fatigue maximal PO and as such is plotted relative to the pre-fatigue maximal PO. This method has been used previously to examine muscle fatigability [9,16,17]. The ability of each muscle to recover from fatigue was monitored for 30 min immediately following the fatigue run. Every 10-min, one set of four work loop cycles were performed and net work was recorded and compared to the pre-fatigue maximal PO [16,17].

The experimental protocol for each muscle was ~190 min. Muscle performance prior to the fatigue run declined by 10 ± 2% (S.E.M), indicating that the quality of all muscle preparations was well maintained throughout the experimental protocol as with similar studies utilising this methodological approach [25].

### 2.4. Muscle Morphology and Calculations

At the end of the experiment, each muscle was detached from the rig, tendons removed and blotted with tissue paper to remove excess fluid. The muscle was weighed using an electronic balance (TL-64, Denver Instrument Company, Arvada, CO, USA) in order to determine wet muscle mass. Mean muscle cross-sectional area (CSA) was calculated from muscle mass, *L*_0_ and an assumed muscle density of 1016 kg∙m^−3^ [32]. Maximal tetanic stress (kN∙m^2^) was calculated as peak tetanic force divided by mean muscle CSA. Absolute PO (Watts) was calculated as the product of net work and CF and was normalised to muscle mass (W∙kg^−1^ muscle mass) at each CF by dividing absolute PO by muscle mass. Muscle mass, muscle length, muscle CSA and absolute force and power for the diaphragm are not reported due to the slight variations in diaphragm size and mass, despite attempting to remove the same section for each animal. Whilst much of this variation in isolated diaphragm preparation size could be natural, only normalised force and power are presented for the diaphragm as per many previous studies [9,26,33,34,35].

### 2.5. Statistical Analysis of Data

All data are presented as mean ± S.E.M. The level of significance was set at *P* < 0.05 for all analyses. Initial tests for normality and homogeneity were performed to determine the appropriate statistical analyses. Differences in animal anthropometrics and isometric properties between the control and HFD groups were measured using an independent *t*-test (Excel 2016, Microsoft, Redmond, WA, USA). Comparisons of the absolute and normalised power output-cycle frequency data were assessed using two-way analysis of variance (ANOVA) using SPSS v.23 (IBM SPSS Statistics for Windows, IBM Corp, Armonk, NY, USA), with diet and cycle frequency as factors. Correlations between animal body mass and maximal normalised work loop power output were performed to determine whether larger animals generated lower power relative to muscle size. Correlations were performed in Excel.

Independent samples *t*-test were used to determine significant differences in time taken to reach <50% of the pre-fatigue maximal PO for all muscles of each dietary group. Recovery was assessed by a two-factor ANOVA with time and diet as the factors. An independent samples *t*-test was used to determine whether there was a significant difference between diet groups in PO normalised to muscle mass after 30-min of recovery.

Effect size (ES) was calculated to determine the magnitude of the effect of the HFD on animal morphology and contractile performance in comparison to the control animals. Cohen’s *d* was calculated and corrected for bias using Hedge’s *g* due to the small sample sizes for each experimental group [36]. Thresholds for interpreting the standardized effect (ES) were as determined as: <0.2 trivial, 0.2–0.6 small, 0.6–1.2 moderate, and >1.2 large [37].

The truncated product method [38] was used to analyse the distribution of *p*-values to provide a *p*-value for each group of multiple hypothesis tests to assess whether these values were biased via multiple hypothesis testing. The truncated product method *p*-value was <0.001, demonstrating that the results were not biased based on multiple hypothesis testing.

## 3. Results

### 3.1. Morphology

The HFD diet group had 24% greater body mass, 21% greater body circumference and 119% greater gonadal fat pad mass than the control group (Table 1; *P* < 0.001). Furthermore, the HFD group nasoanal length, Lee Index of Obesity and Body Mass Index was 5%, 2% and 13% greater respectively (Table 1; *P* ≤ 0.04). The HFD group had 16% greater soleus muscle mass and 18% greater EDL muscle mass (Table 2; *P* < 0.04), and soleus and EDL muscle CSA was 13% and 17% greater in the HFD group than the control group (Table 2; *P* ≤ 0.05). However, soleus and EDL muscle length were not significantly different between each group (Table 2; *P* > 0.30). The FPM of the HFD group accounted for a greater percentage of the total BM than the control group (Table 1; *P* < 0.001). However, when soleus and EDL muscle mass was expressed as a ratio to animal body mass, there were no significant differences between the control and HFD groups for either muscle (Table 2; *P* > 0.66).

### 3.2. Isometric Properties

Maximal absolute tetanus force and maximal tetanus stress were unaffected by a HFD for soleus and EDL (Figure 1A–D; *P* > 0.21, ES = 0.18–0.54). However, there was a tendency for isometric stress to be lower for the HFD group than the control group (Figure 1E; *P* = 0.08, ES = 0.78). There were no significant differences between the controls and HFD groups in tetanus activation and relaxation times (Table 3; *P* > 0.12).

### 3.3. Work Loop Power Output

Absolute PO of the soleus and EDL was significantly higher in the HFD group than the control group, increasing on average by 13% (Figure 2A; *P* = 0.01, ES = 0.36–0.65) and 15% (Figure 2C; *P* = 0.04; ES = 0.28–0.39) respectively. Whilst there was a significant change in optimal CF for PO in HFD soleus (Figure 2A; *P* < 0.001), this was not the case for EDL (Figure 2C; *P* > 0.16). There was no diet*CF interaction for either muscle (*P* = 1.00). When PO was normalised to muscle mass, differences were not apparent between control and HFD in soleus or EDL (Figure 2B,D; *P* > 0.62; ES = 0.00–0.24). In contrast to the locomotor muscles, PO normalised to muscle mass for the diaphragm in the HFD group was significantly lower by an average of 27% across all CF’s in comparison to the control group (Figure 2E; *P* < 0.001; ES = 0.84–1.39). CF had a significant effect on normalised and PO for all groups (Figure 2B,D,E; *P* < 0.05). There was no interaction between diet & CF for all muscles indicating no alteration in the shape of the normalised power output-cycle frequency curves between each group (Figure 2B,D,E; *P* = 1.00).

For the obese group, soleus isolated from obese animals that were heavier produced significantly lower maximal normalised work loop PO (Figure 3B *r* = −0.568, *P* = 0.012) however, there was no significant relationship between maximal normalised PO and body mass for the EDL (Figure 3C; *r* = 0.004, *P* = 0.85). There was a tendency for obese diaphragm isolated from heavier animals to produce lower normalised PO (Figure 3B; *r* = −0.316, *P* = 0.08).

### 3.4. Work Loop Shapes for the Diaphragm

As there were no differences in maximal PO normalised to muscle mass between control and HFD soleus and EDL (Figure 2B,D), work loop shapes were not examined for these muscles.

Maximal work loop PO normalised to muscle mass occurred at 7 Hz for the diaphragm of the control and HFD groups (Figure 2E), with a strain amplitude of ±4% of optimal length typically eliciting maximal net work (Figure 4). The typical work loop shapes at this CF indicated that peak force production was not significantly affected by diet (Control, 65.5 ± 6.0 mN; HFD, 58.7 ± 5.1 mN; *P* > 0.54), however, force during muscle shortening was typically lower for the HFD diaphragm. Additionally, the passive work during lengthening and re-lengthening appears to be greater for the HFD diaphragm than the control, increasing the work done on the muscle to lengthen it, consequently decreasing the net work (Figure 4).

### 3.5. Fatigability and Recovery

Fifty consecutive work loop cycles resulted in a significant reduction in PO, over time for all muscles (Figure 5A,C,E; *P* < 0.0001). However, diet did not significantly affect the time-course of fatigue for each muscle (Figure 5A,C,E; *P* > 0.29), nor time to reach 50% of the pre-fatigue maximum for all muscles (Figure 5A,C,E; *P* > 0.39; ES = 0.05–0.37).

Whilst diet did not significantly impair the ability of soleus and EDL to recover from the fatigue protocol (Figure 5B,D; *P* > 0.27; ES = 0.01–0.11), diet significantly reduced the recovery of PO for the diaphragm from HFD individuals when compared to controls (Figure 5F; *P* = 0.01) where PO was significantly different after 30 min of recovery (Figure 5F; *P* = 0.02; ES = 0.18). There was no time effect on the recovery of PO for all three muscles (*P* > 0.17), nor was a diet*time interaction observed (*P* > 0.37).

## 4. Discussion

Our results indicate that, at the muscular level, dietary-induced obesity in old age significantly impairs respiratory, but not locomotory, isolated skeletal muscle function. Obesity significantly reduced normalised diaphragm PO of old mice following a HFD in comparison to age-matched control animals, which is likely to be related to impaired force generation during muscle shortening and increased passive work through lengthening. By contrast, absolute PO for the HFD soleus and EDL improved compared to age-matched controls, however, this did not alter muscle quality (force/power relative to muscle size). These results differ to findings from younger models of obesity induced via diet which have investigated isolated muscle performance where, locomotory muscle contractile performance is generally poorer [9,21,22], and as such will likely lead to differing consequences for old obese adults during locomotion and respiration.

### 4.1. Age-Related Changes in Skeletal Muscle Contractile Function

When comparing the differences in contractile performance between young CD-1 mice in previous studies and 79-week-old CD-1 mice in the present study, a substantial age-related reduction in contractile performance can be observed for all skeletal muscles examined. Isometric stress for 10-week-old CD-1 females ranges from 199 kN∙m^2^–280 kN∙m^2^ for soleus [16,17,31,39,40,41,42], 233 kN∙m^2^–332 kN∙m^2^ for EDL [16,17,31,39,42], and 155 kN∙m^2^–169 kN∙m^2^ for diaphragm [16,40], whilst PO normalised to muscle mass ranges from 31 W∙kg^−1^–34 W∙kg^−1^ at 5 Hz for soleus [17,31,39,40,41,42], 85 W∙kg^−1^–107 W∙kg^−1^ at 10 Hz for EDL [16,17,31,39], and 47 W∙kg^−1^–60 W∙kg^−1^ at 7 Hz for diaphragm [16,40]. Compared to the aforementioned studies, a substantial decline in isometric stress and normalised PO in this study is observed by 79 weeks of age for control soleus, EDL and diaphragm muscles. (Figure 1 and Figure 2B,D,E).

### 4.2. Effects of Dietary-Induced Obesity on Animal and Muscle Morphology in Old Age

Provision of a calorie-rich diet resulted in the excessive accumulation of gonadal fat and elevated skeletal muscle mass, which ultimately contributed to an increase in animal morphology and body mass (Table 1 and Table 2). The increase in soleus and EDL muscle mass may in part be due to the ectopic accumulation of fat within the muscle [43], although the added load upon the locomotor skeletal muscles in the HFD group may stimulate a hypertrophic effect [44]. Bott et al. [23] reported a hypertrophic effect of 33-week-old C57BL/6J soleus type I, type IIa, type IIx and type IIb fiber CSA in line with an elevated body mass for their HFD group compared to baseline measures and the age-matched control group, which is unsurprising given the postural position of the soleus, though it is interesting to note the comparative fibre atrophy of the non-weight bearing EDL in their study following an obesogenic diet.

### 4.3. Effect of Dietary-Induced Obesity on Isometric Force and Work Loop Power in Old Age

The increased absolute PO of the soleus (Figure 2A) muscle aligns with previous in vivo work demonstrating an increased maximal force of “antigravity”, weight-bearing muscles in older obese adults [3,4,6,7,11]. The increase in absolute PO for the EDL of the HFD group (Figure 2C) is surprising given in vivo work demonstrates little change in absolute force for non-weight-bearing skeletal muscles of old obese adults [3,7]. An added load via the increase in adipose tissue may be a sufficient stimulus to promote a training adaptation, and potentially a hypertrophic effect for the locomotor skeletal muscles that may account for an increased muscle mass and increase in absolute PO of locomotor muscles [9,44]. This seemingly positive response to obesity, however, will have serious in vivo implications due to the elevated muscle mass and fat-free mass contributing to an already elevated body mass. Furthermore, the magnitude of the increase in body mass (24%) is not reciprocated by a similarly proportioned increase in absolute PO for the soleus (13%) and EDL (15%), meaning the ability to overcome a greater bodily inertia will require greater muscular effort. We have shown that, for the soleus, despite unchanged isometric stress and an increase in absolute PO and maintenance of normalised work loop PO in the HFD group, there is a negative association between animal body mass and maximal normalised PO which is not evident for the EDL (Figure 3C,D). This muscle-specific effect for the soleus would mean in vivo muscle stabilisation at the ankle [31] and postural control in older obese adults may be significantly impacted, leading to a poorer capacity to perform activities of daily living compounded by a poorer gait, slower speed of performing activities, and lower fatigue resistance [45].

By contrast, a HFD in older animals caused a significant reduction in diaphragm PO relative to muscle mass (Figure 2E). As with locomotor muscle, fat is likely to be stored ectopically within the diaphragm, increasing the non-contractile mass and work required to lengthen the muscle. However, unlike locomotor muscles, adipose tissue loading on the diaphragm is unlikely to induce hypertrophy due to elevated adipose tissue in the thoracic cavity of obese adults increasing diaphragm compliance, and thus increases respiratory resistance [46,47]. In terms of the work loop, a greater non-contractile mass and lower tissue compliance would amplify the work required (negative work) to lengthen the muscle, and would, therefore, decrease maximal net work and PO [24]. The impairment in PO does not appear to be limited by the ability for old, obese diaphragm to produce peak force during cyclical work, but instead, maintenance of force during shortening is lower, with a tendency for greater eccentric (i.e., negative) work during re-lengthening compared to control animals (Figure 4). Consequently, it is plausible that a reduced capacity for old obese diaphragm to generate PO to be a contributor to the increased metabolic and cardiovascular disease risk in old obese adults [48].

### 4.4. Fatigability and Recovery

Dietary-induced obesity did not cause a significant reduction in the ability to sustain PO over repeated work loops for old obese soleus, EDL or diaphragm (Figure 5A,C,E), nor for locomotor muscles to recover from the fatigue protocol (Figure 5B,D) as found with young obese female mice [14]. Whilst the pattern of fatigue appears the same for both the control and HFD diaphragm, each data set is plotted as a percentage of the pre-fatigue maximal PO. Power normalised to muscle mass is significantly lower in the HFD group, so it should be considered that PO at 100% for the HFD diaphragm group would be significantly lower due to a lower starting normalised PO, and therefore would be likely to fatigue faster in vivo when working at a comparable intensity to the control diaphragm. Recovery of old obese diaphragm is significantly impaired after 30 min of recovery, but not 10 or 20 min (Figure 5F), despite no change in the fatigue response. The pattern of fatigue and recovery is fibre-type dependent, where slower fibres are more fatigue resistant and exhibit faster recovery following fatigue. The patterns of fatigue and recovery in this study are similar to our previous work to have utilised this fatigue recovery protocol [16,17]. Ageing and obesity are associated with a shift in fibre type, with ageing associated with a fast-to-slow fibre shift, whilst the change in fibre type composition in obesity is equivocal [2]. It is unlikely that fibre shifting has occurred for any of the skeletal muscles in this study, given that fatigue resistance (Figure 5A,C,E), isometric activation and relaxation times (Table 3) were unchanged in the HFD groups compared to controls. Moreover, the pattern of recovery does not indicate a fibre shift towards a fast fibre type, as both soleus and diaphragm recover well and relatively quickly following the fatigue protocol as with our previous work [16,17]. For the soleus and diaphragm, power recovers to within 90% of the pre-fatigue maximal 10 min following completion of the fatigue protocol and is well maintained across the recovery period (Figure 5B,F). Poorer recovery of the obese diaphragm after 30 min may be due to a small amount of fibre damage caused by the fatigue protocol, resulting in further development of an anoxic core, which has been shown to impair contractile performance during the course of experiments using isolated skeletal muscles [49]. In addition, the pattern of fatigue and recovery of control and HFD EDL in this study (Figure 5C,D) is similar to that of previous work from our lab examining contractile properties of animals of the same age, sex and strain [17].

The added load of an increased bodily inertia in older adults could be a significant contributor towards reduced muscular endurance in older adults [50]. It is expected that the added muscle mass and fat-free mass in old obese adults is likely to further contribute to a reduction in fatigue resistance when working at the same relative intensities due to isolated skeletal muscles fatiguing at the same rate with no change in muscle quality [9]. As such, it is more likely the increased demand placed on the muscle due to an elevated body mass, rather than the ability of the skeletal muscle to withstand fatigue, may potentially explain the reduction in whole animal exercise tolerance following a HFD [22] and slower gait velocity in old obese adults [51].

### 4.5. Differences in Contractile Performance between Young and Old Models of Obesity

Whilst obesity studies in young rodents share similar characteristics with the present study, such as inducing obesity via diet, and comparing the soleus and EDL to represent phenotypic differences, comparisons are difficult due to the different methodological approaches including feeding duration and diet composition, a lack of classification of what is considered obese for rodent models, and different test temperature for isolated skeletal muscles [2].

For soleus, absolute force, isometric stress, and absolute and normalised PO remain unchanged or even improve following an obesogenic diet in young female CD-1 mice fed the same HFD [9] which does not differ to the current findings. The EDL response, however, is more ambiguous. Whilst isometric force and stress was well maintained in old obese EDL in the present study, some previous studies report a reduction in isometric force and stress in young obese EDL [22], whereas others report no change in isometric stress [21,23,26]. The increase in absolute PO for old obese EDL is a surprising adaptation, considering that obesity in younger rodents does not promote a positive adaptation for this muscle, particularly given that obesity may cause fibre atrophy for the EDL [23]. The lower diaphragm stress and normalised PO of old obese mice mirror that of young obese mice [9,26]. It is possible that obesity has a phenotypic effect on skeletal muscle fibres, where the contractile function of type II and IIa/x fibres are affected to a greater extent than type I fibres, which may explain the muscle-specific differences between young and old obese contractile performance. With age, muscle composed of predominantly fast-twitch muscle fibres experience shifting towards a slow-twitch composition, which consequently may be less affected by obesity than fast-twitch fibres, hence no change in muscle quality for old EDL and soleus. To date, no study has examined the effects of obesity on single fibre contractile function to determine whether obesity has a phenotypic response. In lieu of the present findings, obesity exacerbates the ageing process for the diaphragm only when contextualized with respect to younger animals in our previous work [9,26]. Even at a young age, a HFD causes the greatest decline in muscle quality for the diaphragm, where work loop PO of 30-week-old diaphragm of animals fed a HFD for 16 weeks is poorer than that of 79-week-old diaphragm in this study in which animals were not fed a HFD [9]. Further investigation is required to determine whether impaired diaphragm contractility is comparable between young and old obese humans, and to what extent respiratory function is affected.

### 4.6. Limitations and Future Directions

We have outlined the limitations of normalising contractile performance to muscle mass in detail in our previous work [9,16,17]. Whilst isometric stress and normalized work loop PO provide an accurate assessment of muscle quality, it is considered that in old obese skeletal muscles, a smaller proportion of the total mass will be contractile protein due to the greater infiltration and accumulation of lipids. It would be of greater importance to normalise contractile performance to lean tissue mass to further consider to what extent lipid accumulation impacts on changes in muscle quality. However, there are significant methodological problems with accurately obtaining measure of intramuscular lipid and contractile mass, where previous work has indicated that obesity can cause significant increases in skeletal muscle lipid content in a muscle-specific manner [52]. Considering muscle is denser than fat, the contribution of muscle lipids to a significant increase in muscle mass of the HFD muscles in this study is likely to be small, and for that reason, muscular lipids will play a small role in changes in muscle quality.

The onset of obesity does not suddenly occur in old age, rather an old obese adults’ high body fat content is more likely due to a prolonged duration of poor dietary choices, which may alter body composition and muscle contractile function in a muscle-specific and sex-specific manner. However, the present study has given further insight into the relative effects of aging and obesity in old mice. Other studies have indicated that there is further complexity in the interaction between obesity and aging. For example, DeNies et al. [14] reported that 52 weeks of a HFD caused a significant reduction in type I fibres for the soleus of male, but not female, C57BL/6J mice when compared to animals fed a control diet. Moreover, obese females had significantly more type I fibres for the soleus compared to males fed a HFD. These findings indicate that prolonged consumption of a HFD affects skeletal muscle fibre type distribution for in a sex-specific and muscle-specific manner, which may lead to poorer contractility parameters such as isometric stress, normalised power, and fatigue resistance following a prolonged diet which is likely to be exacerbated in males compared with females. However, the relationship between, sex and obesity in young and old rodents in relation to contractile function and muscle morphology has yet to be fully explored, following a short duration and prolonged duration of a HFD. Each independent factor warrants further exploration via contractile, biochemical and histological measurements to better elucidate the mechanisms that affect skeletal muscle contractile function in obesity models.

The current study utilises an isolated muscle approach to examine the impact of inducing obesity in old age on contractile function of skeletal muscle independent of any potential effects on the central nervous system. It is well documented that denervation of skeletal muscles is an inescapable consequence of the ageing process [53]. The impact of obesity on the neuromuscular junction, however, is poorly understood. Conducting in situ experiments, where the skeletal muscle neuromuscular junction remains intact and is innervated by an external electrical source, could provide a valuable insight into the effect of obesity on contractile function in relation to the neuromuscular junction, and whether this is worsened in old obese skeletal muscles.

The reversibility of obesity in older adults is of importance due to the numerous benefits of physical activity on body composition and quality of life. Rodent models have shown that 8 weeks of voluntary exercise in 38-week-old CD-1 mice can evoke an improvement in soleus and diaphragm stress, normalised PO and body mass compared to untrained age-matched controls [52]. As animal body mass increases with age, and substantially more so following a HFD, it is possible that voluntary wheel running can promote an improvement in weight status and contractile performance in older mice, and potentially reverse the negative effects of obesity in older animals in general.

## 5. Conclusions

Absolute force and PO, muscle quality, and fatigability of locomotor muscles is well maintained following a HFD in old animals. By contrast, muscle quality of old obese diaphragm is significantly lower compared to lean, age-matched counterparts. These findings differ to that of young obese skeletal muscles, where both force and PO normalised to muscle mass generally declines for EDL and diaphragm, though soleus muscle quality is well maintained irrespective of age. An elevated body mass in old obese adults is likely to act as a training stimulus on the soleus and EDL, as demonstrated by an increase in the absolute PO for the locomotor muscles. However, the increased bodily inertia will mean acute and sustained in vivo locomotor performance is likely to be substantially affected due to a larger limb mass and body mass creating a greater work demand on the skeletal muscle. The elevated fat mass loaded on the diaphragm in vivo could be a plausible contributor explaining the reduction in muscle quality as found in vitro. This may potentially elevate respiratory disease risk and further contribute to the negative cycle of obesity.

## Figures and Tables

**Figure 1 nutrients-11-00505-f001:**
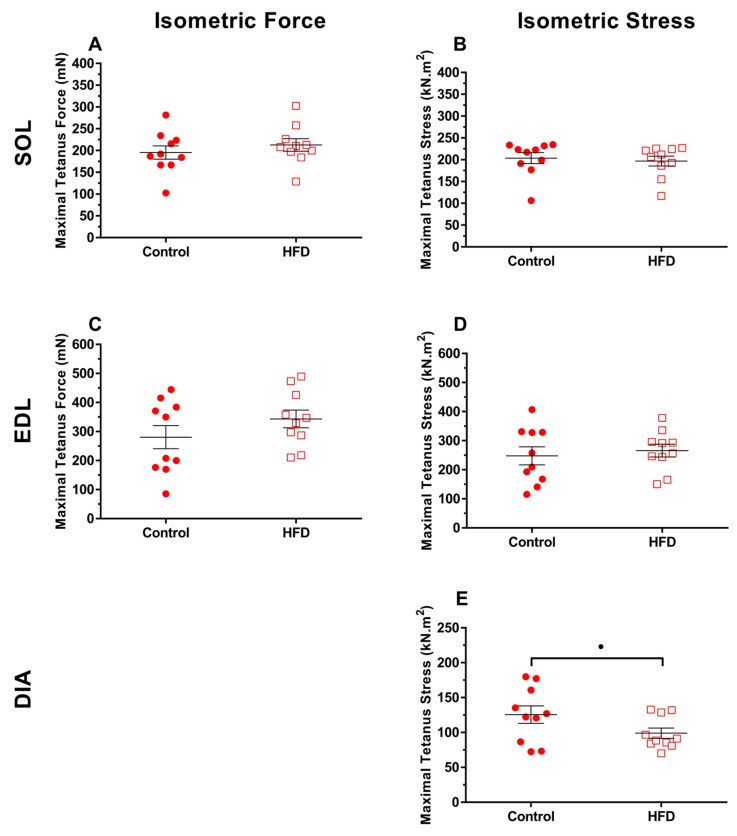
The effect of 9 weeks of a high-fat diet (HFD) on the absolute maximal isometric force (**A**,**C**) and maximal isometric stress (**B**,**D**,**E**) of isolated mouse soleus (**A**,**B**), EDL (**C**,**D**) and diaphragm (**E**) from 79-week-old mice. *n* = 10 per muscle per group. Absolute force is not presented for diaphragm due to slight differences in the segment of the diaphragm taken for each experiment. The • symbol indicates a statistical tendency (*P* = 0.08) for isometric stress to be lower in the HFD group (ES = 0.78).

**Figure 2 nutrients-11-00505-f002:**
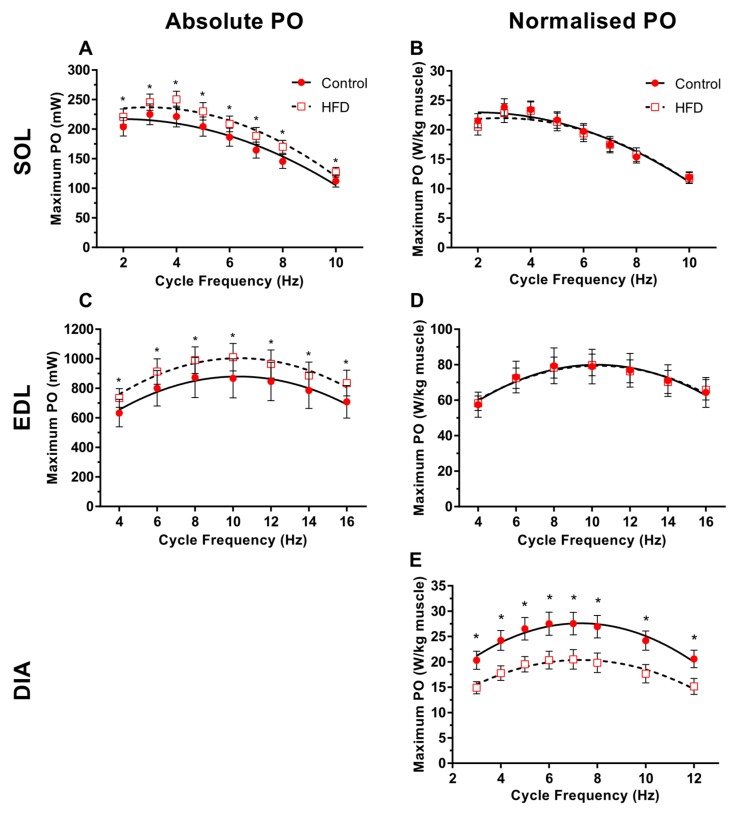
The effect of 9 weeks of a high-fat diet (HFD) on the power output-cycle frequency relationship for absolute power output (PO) (milliWatts; **A**,**C**) and normalised PO (Watts per kilogram of muscle mass; **B**,**D**,**E**) of isolated mouse soleus (**A**,**B**), EDL (**C**,**D**) and diaphragm (**E**) for the control or HFD groups. *n* = 10 per muscle. Absolute power is not presented for diaphragm due to slightly different segments of the diaphragm being used for each experiment. A * symbol denotes a significant difference in power output between each experimental group at a given cycle frequency.

**Figure 3 nutrients-11-00505-f003:**
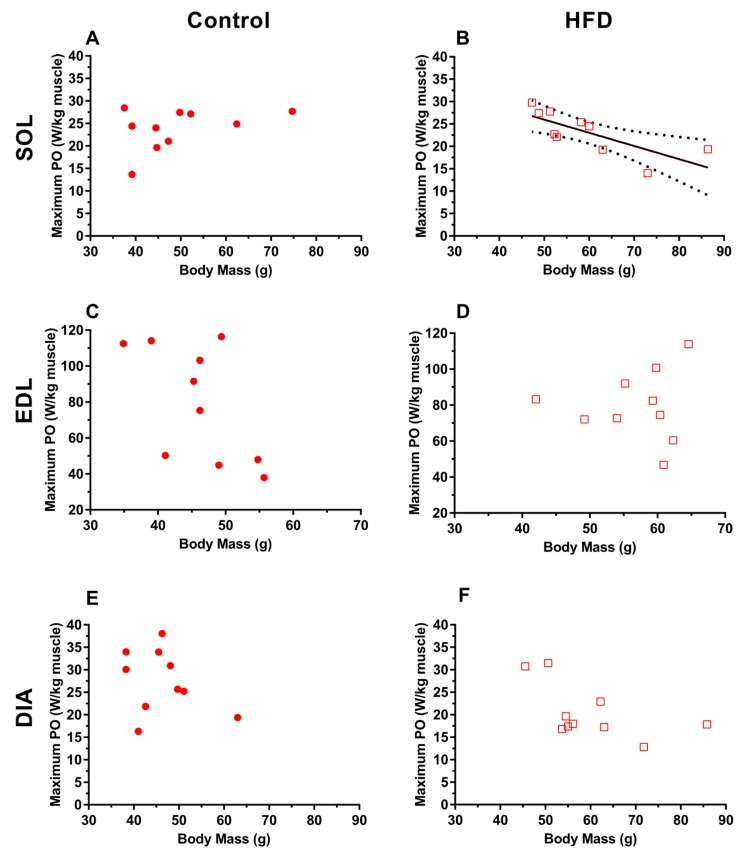
The relationship between whole animal body mass and normalized work loop power for control (**A**,**C**,**E**) and HFD (**B**,**D**,**F**) soleus (**A**,**B**), EDL (**C**,**D**) and diaphragm (**E**,**F**) experimental groups. *n* = 10 per muscle per group. For figure (**B**) the lines represent a first-order polynomial fitted to the data using a least squares regression and the 95% confidence limits of this line.

**Figure 4 nutrients-11-00505-f004:**
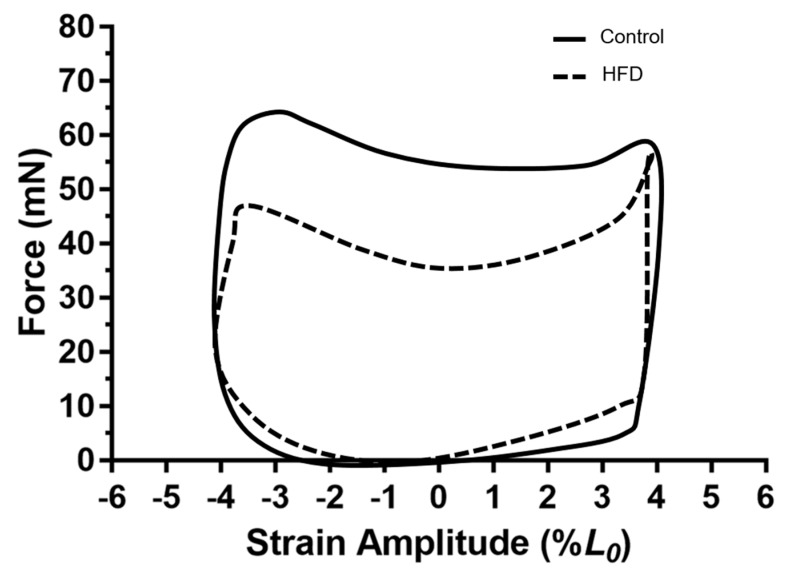
The effect of dietary-induced obesity on 79-week-old mice on diaphragm work loop shapes for the control (solid) or HFD (dashed) groups at a cycle frequency of 7 Hz, where maximal power output was elicited. Figures plotted as force against strain amplitude (%*L*_0_). The third work loop of the set of four work loop stimulations is shown for each group. Work loops interpreted in the anticlockwise direction from 0 of strain amplitude.

**Figure 5 nutrients-11-00505-f005:**
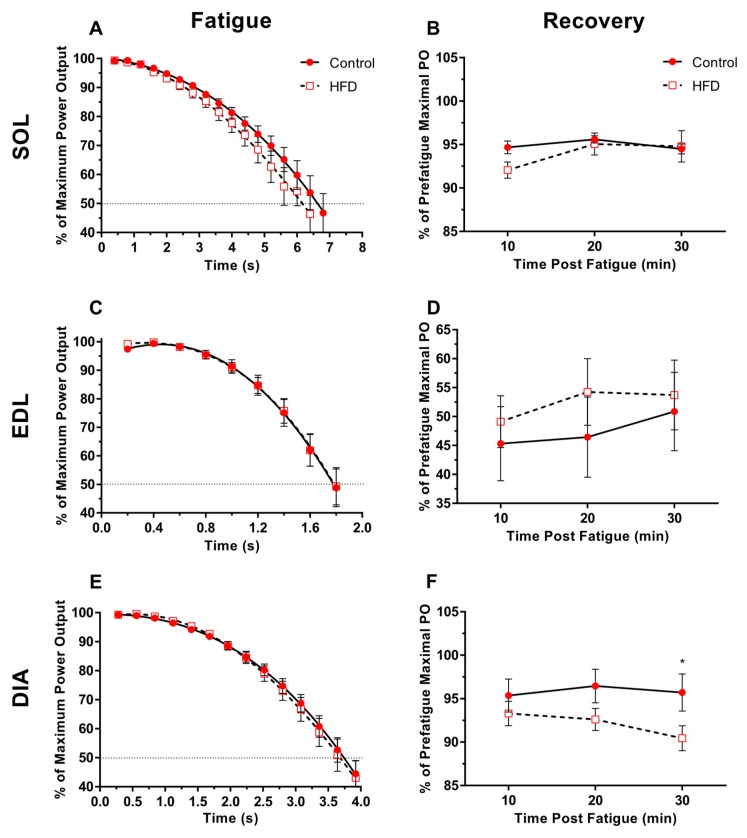
The effect of 9 weeks of a HFD on the fatigue resistance (**A**,**C**,**E**) and recovery of power (**B**,**D**,**F**) of maximally stimulated mouse soleus (**A**,**B**), EDL (**C**,**D**) and diaphragm (**E**,**F**) for the control and HFD groups. Values presented as mean ± S.E.M. A * denotes a significant (*P* < 0.05) difference in recovery of power between experimental groups at a given time point.

**Table 1 nutrients-11-00505-t001:** The effects of 9 weeks of a high-fat diet (HFD) on animal anthropometrics for 79-week-old mice.

	Control	HFD	*p*-Value	Effect Size
Body Mass (g)	47.2 ± 3.0	58.6 ± 3.6	<0.001	1.19
Nasoanal Length (cm)	11.8 ± 0.2	12.4 ± 0.2	<0.001	0.91
Body Circumference (cm)	8.4 ± 0.4	10.6 ± 0.6	<0.001	1.26
Body Mass Index (kg∙m^2^)	0.34 ± 0.01	0.38 ± 0.02	<0.001	0.87
Lee Index of Obesity	305 ± 5	313 ± 5	0.04	0.54
Fat Pad Mass (g)	3.6 ± 0.9	7.9 ± 1.2	<0.001	1.23
Fat Pad Mass:Body Mass (%)	7.0 ± 0.8	12.8 ± 0.8	<0.001	1.32

Values presented as mean ± S.E.M; *n* = 30 control; *n* = 30 high-fat diet.

**Table 2 nutrients-11-00505-t002:** The effects of 9 weeks of a high-fat diet (HFD) on the muscle-specific morphology for 79-week-old mice.

	Soleus				EDL			
	Control	HFD	*p*-Value	Effect Size	Control	HFD	*p*-Value	Effect Size
Muscle Mass (mg)	9.4 ± 0.5	11.0 ± 0.6	0.04	1.02	10.6 ± 0.6	12.6 ± 0.5	0.014	1.19
Muscle Length (mm)	9.3 ± 0.1	9.5 ± 0.2	0.30	0.38	9.1 ± 0.2	9.1 ± 0.2	0.76	0.00
Muscle CSA (m^2^)	1.0 × 10^−6^ ± 4.5 × 10^−8^	1.1 × 10^−6^ ± 3.9 × 10^−8^	0.05	0.89	1.1 × 10^−6^ ± 5.9 × 10^−8^	1.3 × 10^−6^ ± 4.2 × 10^−8^	0.02	1.12
Muscle Mass:Body Mass (%)	0.52 ± 0.04	0.55 ± 0.03	0.67	0.25	0.45 ± 0.04	0.45 ± 0.02	0.98	0.00

Values presented as mean ± S.E.M; *n* = 10 per muscle per group. Data not presented for diaphragm as morphological comparisons cannot be made due to different sections of the diaphragm isolated during each preparation. CSA, cross-sectional area.

**Table 3 nutrients-11-00505-t003:** The effects of 9 weeks of a high-fat diet (HFD) on isometric activation (THPT) and relaxation (LSHR) of isolated mouse soleus, EDL and diaphragm from 79-week-old mice.

	THPT (ms)				LSHR (ms)			
	Control	HFD	*p*-Value	Effect Size	Control	HFD	*p*-Value	Effect Size
Soleus	37.6 ± 2.3	40.1 ± 2.2	0.38	0.38	52.3 ± 3.2	48.0 ± 1.9	0.21	0.55
EDL	16.0 ± 1.1	16.3 ± 0.9	0.81	0.10	17.4 ± 1.2	17.6 ± 1.1	0.89	0.06
Diaphragm	24.4 ± 1.0	26.5 ± 1.2	0.21	0.60	25.0 ± 1.0	26.8 ± 0.8	0.13	0.69

Values presented as mean ± S.E.M. *n* = 10 for each muscle of each group. THPT, time to half-peak tetanus; LSHR, last stimulus to half tetanus relaxation.

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
