# Peer review of "Does Dietary-Induced Obesity in Old Age Impair the Contractile Performance of Isolated Mouse Soleus, Extensor Digitorum Longus and Diaphragm Skeletal Muscles?"

_nutrients, 2019, doi:10.3390/nu11030505_

Round 1

Reviewer 1 Report

Cameron Hill and coworkers tackled to reveal the effect of the ageing and the obesity on muscle functions. They measured physiological muscle contraction properties using isolated-muscles from mice. Unfortunately, most of their data showed no significant combination effect of the ageing and the obesity on muscle functions regardless the authors’ effort. However, one of the most important points which the authors found is that recovery of old obese diaphragm was significantly impaired. The authors described their thoughts about impaired recovery of diaphragm in the Discussion section but rather descriptive. The authors should dig a bit deeper to explain this finding from the different angles such as the histological point. I would recommend that the authors should conduct some experiments to explain the malfunction of diaphragm muscle recovery in old obese mice.

Major

Muscle morphology

Although the authors show muscle morphology data in Table 2, these were descriptive. Some histological data would be more powerful to compare properties of each muscle. Since HE stained-muscle sections can directly show the size of muscle fibers and the ectopic fat deposition. Antibodies against myosin heavy chain can tell the difference of muscle fiber type among different muscles (see below). I would recommend that the authors carry out histological works in order to improve the authors’ manuscript.

Muscle fiber type

Muscle fiber type composition is different among EDL, soleus and diaphragm muscles. I am wondering whether high fat diet affects the muscle fiber type composition or the fiber type shift. It would be nice to show these data. This might be explained the reason of impaired recovery as the authors found.  

Table 2

The authors described the effects of a high-fat diet on the muscle morphology in Table 2. However, data on diaphragm was not found in Table 2. To add diaphragm data in Table 2 would improve the authors’ manuscript.

Minor

Line 44

What is “Chang”?

Figure 2 and 5

What is the unit of the X-axis?

Author Response

Please see the attached response letter for reviewer one.

Reviewer 2 Report

Major

1.       Authors must be rewrite title, introduction and discussion. There were not the young with and without high-hat diet groups, so authors could not investigate about “age-related” decline skeletal muscle function and “synergistic effects of aging and obesity” on skeletal muscle function. The things what authors can investigate are that the effects of high-fat diet on isolated skeletal muscle function in old aged mice. If authors tend to investigate “age-related” decline and “synergistic effects of aging and obesity” on skeletal muscle function, I think that design in this study must be reconsidered.

Minor

1.       Line44

What is “chang”?

2.       Line186

What is “L0”?

3.       Table 1

In Body mass in control, 3x.0 may be 3.0.

4.       Table 2

Mean values of muscle CSA in the High-Fat Diet of both Soleus and EDL, and the Control of EDL must be expressed as 10-7. Authors should not use different power in the same place.

5.       What are x axis in the Figure 2 and 5.

6.       There are not graphs of DIA in Figure 1 and 2.

Author Response

Please see attached the response to reviewer two.

Reviewer 3 Report

The manuscript by Hills C et al., describes that Does Dietary-Induced Fatness Exacerbate the Age-2 Related Decline in Isolated Mouse Soleus, EDL and 3 Diaphragm Skeletal Muscle Contractile Function? This is an interesting possibility and likely to be potentially important information for the molecular nutrition of muscle. In addition, this manuscript could be compared with your previous report (Tallis J et al., J Appl Physiol (1985). 2017 Jan 1;122(1):170-181) regarding the information of young obese mice. This is a carefully done study and the findings are of considerable interest. A few minor revisions are listed below.

Previous your report has been shown that the data of changes in myosin heavy chain expression and AMP-activated protein kinase activity. In this case, did these makers change in elder obese mice?

The authors used female mice in this experiment. Is there a difference in gender?

As you know, Age-dependent declines in muscle function are involved in the alteration of neuromuscular junction (NMJ). In this experiment, Does daietary-induced fatness affect the age-related disturbance in NMJ?

In Figure 2 and 5, the authors should describe that what does the horizontal axis means.

5.    line 281, MnàmN.

Author Response

Please see attached the response to reviewer three.

Round 2

Reviewer 1 Report

The manuscript has been improved.

Author Response

Please see attached the response to reviewer one.

Reviewer 2 Report

Authors want to compare between young and old, the young groups are needed.

Author Response

Please see attached the response to the comment provided by reviewer two.

Round 3

Reviewer 2 Report

I think that authors focus on the effect of high-fat diet in old rats and should not compare between young and old, because authors had the young groups even if authors previously performed the experiments using the young groups.

Author Response

Please see attached the authors' response to reviewer two.
